# Smart City and High-Tech Urban Interventions Targeting Human Health: An Equity-Focused Systematic Review

**DOI:** 10.3390/ijerph17072325

**Published:** 2020-03-30

**Authors:** Adrian Buttazzoni, Marta Veenhof, Leia Minaker

**Affiliations:** 1School of Planning, Faculty of Environment, University of Waterloo, Waterloo, ON N2L 3G1, Canada; anbuttazzoni@uwaterloo.ca (A.B.); marta.veenhof@uwaterloo.ca (M.V.); 2Geographies of Health in Place, Planning, and Public Health Lab, Faculty of Environment, University of Waterloo, Waterloo, ON N2L 3G1, Canada; 3School of Public Health and Health Systems, Faculty of Applied Health Sciences, University of Waterloo, Waterloo, ON N2L 3G1, Canada; 4University Avenue West, Environment Building 3, Waterloo, ON N2L 3G1, Canada

**Keywords:** built environment, equity, interventions, review, smart cities, urban health

## Abstract

Urban infrastructure systems responsible for the provision of energy, transportation, shelter, and communication to populations are important determinants of health and health equity. The term “smart city” has been used synonymously with other terms, such as “digital city”, “sustainable city”, and “information city”, even though definitional distinctions exist between terms. In this review, we use “smart cities” as a catch-all term to refer to an emerging concept in urban governance practice and scholarship that has been increasingly applied to achieve public health aims. The objective of this systematic review was to document and analyze the inclusion of equity considerations and dimensions (i.e., a measurement, analytical, or dialectical focus on systematic disparities in health between groups) in smart city interventions aimed to improve human health and well-being. Systematic searches were carried out in the Cumulative Index to Nursing and Allied Health Literature (CINAHL), Psychological Information Database (PsycINFO), the PubMed database from the National Center for Biotechnology Information, Elsevier’s database Scopus, and Web of Science, returning 3219 titles. Ultimately, 28 articles were retained, assessed, and coded for their inclusion of equity characteristics using the Cochrane PROGRESS-Plus tool (referring to (P) place of residence, (R) race, (O) occupation, (G) gender, (R) religion, (E) education, (S) socio-economic status (SES), and (S) social capital). The most frequently included equity considerations in smart city health interventions were place of residence, SES, social capital, and personal characteristics; conversely, occupation, gender or sex, religion, race, ethnicity, culture, language, and education characteristics were comparatively less featured in such interventions. Overall, it appears that most of intervention evaluations assessed in this review are still in the early testing phases, and thus did not include or feature robust evaluative designs or commercially available technologies

## 1. Introduction

Urban infrastructure—including systems that provide energy, transportation, shelter, and communication—is an important determinant of population health and health equity. These systems influence risk of morbidity and mortality from injury, and chronic physical and mental diseases. Relevant health risks include exposure to traffic, air pollution, noise, and social isolation; sedentary behavior; and unhealthy food sources [1]. These risks are inequitably distributed in cities—disadvantaged groups have the highest rates of exposure to, and morbidity and mortality from, these risks [2]. Cities produce health inequities that are “systematic, socially produced (and therefore modifiable), and unfair” [2]. Of note, the world is undergoing rapid urbanization—an estimated 60% of urban areas that will exist in 2050 are yet to be built [3]. Over the next three decades, 2.5 billion more people will be living in urban areas [4], creating a massive opportunity to build cities that consciously prioritize health and health equity.

The concept of a “smart city” is a new frontier in urban governance that can be applied to achieve public health aims. The term “smart city”—a city that uses information and communication technology (ICT) to improve productivity and achieve more open governance—has been called an “empty signifier” in urban planning [5], as there is no standard definition and the phrase is often void of substantive meaning. Current smart city examples vary substantially in terms of technological maturity, quality of ICT infrastructure, and even smart city objectives [6]. The term smart city is often offered as a panacea for economic prosperity, ecological sustainability, and efficiency, with a common assumption that smart city benefits will be realized equally by all citizens [7]. However, despite idealistic language around using ICT to create sustainable, healthy cities, actual smart city examples rarely consider human health, and few explicitly address governance [7]. Many projects are driven by ICT companies with novel products. Thus, smart city projects have been criticized for being solutions in search of problems [8]. Other critiques include personal privacy risks and cyber security, exacerbation of inequity by concentrating benefits (i.e., advanced infrastructure and services) in specific areas and excluding others, “digital marginalization” (reflecting communities’ connectivity, technology affordability, and capacity to consume information), and inequitable opportunities for public involvement [7]. A new analytical framework for understanding smart cities distinguishes between smart city 1.0 and smart city 2.0, with smart city 2.0 strategies focused on people first and technology as a tool to serve citizens, rather than as an end in itself [9]. Under the smart city 2.0 paradigm, objectives of technological development include mitigating social challenges, enhancing citizen well-being, and addressing citizen needs. These stand in contrast to smart city 1.0 objectives, which include optimization of infrastructure and services, spurring new business opportunities, and addressing universal technical agendas related to energy, transport, and the economy. Finally, the smart city 2.0 approach is decentralized, recognizing contributions from diverse actors, and develops endogenously in response to contextual needs, compared to the centralized approach and exogenous development of smart city 1.0 approaches [9]; ideas which have become staples in many mainstream contemporary practices. One prominent example is the concept of living labs, which is a citizen-oriented approach that is built upon integrating users and other stakeholders and drawing from their experiences and input to inform the development of various projects by reducing risks and improving their acceptance [10].

## 2. History of Smart Cities

While discussions regarding the conceptualization of smart cities began over three decades ago [11], their development can broadly be outlined in the development of two distinct paradigms [9]. In the original paradigm, smart city 1.0, primary emphasis of the approach was placed on technology and data being the most effective means to address prominent planning issues [12]. Smart cities, therefore, represented a top-down approach, which promoted expert-driven issue-framing and problem-solving methods designed to maximize process efficiency [13]. Concurrently, individual residents played a very limited role in smart city 1.0, generally only serving as consumers or end-users in the planning process [9]. Such dismissiveness of the public, coupled with the privileging of special interests and expert opinions, however, eventually inspired extensive criticism targeted at the private sector’s dominance over public values and input in planning processes [12]. Such criticism led to a refinement of the smart city approach.

In the contemporary smart city 2.0 paradigm, a drastic move toward bottom-up [14] and collective urban planning is prioritized and has become a central feature of the approach [15]. Specifically, the new smart city paradigm promotes a planning process where there is increased support for the local citizenry’s capacity to address municipal issues [9], citizens and other non-traditional actors have more opportunity to participate in and contribute to discussions, and the overall structure is oriented to more closely reflecting the needs and preferences of the entire community [16,17,18,19]. As a result smart city 2.0 has brought forth the promotion of more citizen-focused and representative planning processes [20,21,22,23], and has also helped usher in the emergence of more socially conscious research for smart cities.

### 2.1. Smart Cities and Public Health

Planning issues and solutions have the potential to shape community health outcomes. For instance, housing, transportation, social services, and other city-scale issues can profoundly affect the general health and well-being of urban populations [1,24,25,26]. In their 2016 literature review published in the Lancet, Giles–Corti and colleagues [1] demonstrate direct and indirect pathways by which urban and transportation planning affect health. For example, transport policies can influence transport mode outcomes (e.g., proportion of commuters using active transportation modes), which influences population-level risk exposures related to traffic, air pollution, noise, and sedentary behaviors, all of which contribute to health and well-being. Smart city approaches [7] and related technological concepts, such as the Internet of Things (IoT) [27] and mobile health (mHealth) [28], have been suggested to be promising proximal tools and strategies for promoting health and well-being at a population level.

The concept mHealth can be defined as a “public health practice supported by mobile devices, such as mobile phones, patient monitoring devices … and other wireless devices” [29]. While not expressly related to planning, mHealth interventions have been indirectly used to address several urban health issues. Different mHealth strategies have targeted a wide range of health issues, such as improving physical activity via utilizing monitoring, visualization, and digital education [7]; tracking and measuring lifestyle behaviors through employing sensor-based home and body area networks [30]; and building extended healthcare access by running telemedicine programs [31]. Perhaps most importantly, previous systematic reviews have found that mHealth interventions are capable of affecting significant changes in human behavior [28,32,33].

As electronic and mobile tools have continued to coalesce with the continued pursuit of advancing and building healthier lifestyles and environments, recent pushes from the academic community have been made to increase the presence of mHealth strategies in smart city research [34]. On this point, Solanas et al. [34] proffer the concept of “smart health”, succinctly describing that “smart health (sHealth) is the provision of health services by using the context-aware network and sensing infrastructure of smart cities”. Smart and innovative urban planning approaches—or s-health approaches, in this context—have been increasingly linked to achievements of public health objectives, such as improvements in well-being and quality of life [1]. Noting the potential of this approach, the inclusion of specific electronic and mHealth concepts are becoming more commonplace in planning interventions, especially those that clearly intersect with public health. Consequently, more smart city research specifically examining public health topics has emerged, as evidenced by the inclusion of the smart city approach in research areas including healthy ageing [35], food systems [36], and active transport and physical activity [37].

### 2.2. Equity and Smart Cities

Health equity is a worldwide public health objective [38]. Currently, health inequities persist, as evidenced through intergroup differences in the incidence, prevalence, and burden of diseases and mortality, as well as other adverse health conditions [39]. Such disparities are commonly noted in intergroup differences based along gender, race or ethnicity, education or income, physical disability, geographic location, or sexual orientation lines [40]. The achievement of health equity is a desirable goal for all societies, as health disparities are detrimental to all members of a community, not just those who are not members of the privileged class [41]. For example, disparities can lead to the spread of infectious diseases [42] and development of social circumstances that result in upticks in the occurrence of violence and crime [43], which consequently degrade health at a population level. Health equity approaches aim to minimize the influence and impact of such detrimental social circumstances in furtherance of helping all people to reach their full potential [44].

Smart city approaches, as previously noted, have been widely critiqued for their absence of focus regarding social concerns [12,45,46,47]. Scholars have been critical of the perceived dominance of neoliberal economic interests [48,49,50], top-down corporatism [51], and global capital [52] in the priorities of smart city approaches. Accordingly, many arguments have been made suggesting that the approach’s obsession with technological innovation in service of economic growth results in it neglecting health and other social implications [49]. Smart city approaches have also been extensively critiqued for not critically examining the underlying assumptions (i.e., equal access and benefits for all) associated with their proposed technology-based strategies [53], as well as for often lacking a true and grounded understanding of their social environments’ reality [51]. In fact, this critique has been so pervasive it has been coined as “technophilia” (i.e., the love of tech and it featuring as the overwhelming object of study) by critics [54]. Despite these noted shortfalls, recommendations continue to be made that linking smart city concepts with public health and giving honest and thoughtful consideration to social issues can produce noteworthy research and enhance the contribution of smart city scholarship [55].

## 3. Materials and Methods

### 3.1. Current State of Reviews and Justification

Recent reviews regarding smart cities have focused on governance [56], sustainable development [57], and the utilization of open government data [58]; however, equity has yet to be systematically explored. This is, to the authors’ knowledge, the first review to explicitly document and synthesize research on smart city strategies to improve health while specifically focusing on the literature’s consideration and inclusion of equity and social dimensions. In addition to this, there are a few other important points which further justify this review. First, having noted the many criticisms of smart city approaches in consideration (or lack thereof) of equity and social concerns, this review offers a comprehensive assessment and documentation of the progress of several social and justice aspects related to smart cities. Second, the scope of this review captures both qualitative and quantitative work, and also documents other important aspects related to smart city intervention design and methodology. Last, prominent theses such as “splintering urbanism” have argued that since the 1960s, infrastructure networks have become increasingly “unbundled”, leading to fragmentation of the social and material fabric of cities [59]; subsequent scholarship has noted that the underlying neoliberal sources of this trend have weakened equity [60], heightening the importance of assessing modern urban equity. This review emphasizes equity-related methodology and design, principally because we wanted to recognize the areas of study design that have done well in incorporating equity dimensions, to identify those areas of study which should be targeted for improvement in future research, and to promote social equity in the study of urban infrastructure. Lastly, this review utilizes the Cochrane equity tool to generate thematically organized findings, and subsequently a pragmatic discussion for practitioners. The Cochrane equity tool differs from more well-known tools, such as health impact assessments—which tend to focus on broad cultural, economic, and social determinants of health—in two main ways: first, it provides specific definitions of distinct equity characteristics (e.g., education, occupation), and second, it offers and outlines a spectrum of factors in order to guard against suggesting that equity is a single, broad determinant of a systemic health issue [61].

### 3.2. Review Question and Objectives

To guide this review, we posed the following question: What are the main equity approaches, characteristics, features, objectives, and outcomes of smart city interventions that aim to improve human health and well-being? Petticrew and Roberts’ [62] PICOC model - population, intervention, comparison, outcome, and context – for review questions was applied to operationalize and ensure the rigor of this research question. The review question breaks down as follows:

Population: Any group;

Intervention: Smart city intervention aimed at improving human health or well-being 

(as defined below);

Comparison: Any;

Outcome of Interest: Consideration, inclusion, or analysis of sex, gender, socio-economic

Status (SES), race, ethnicity, culture, and disability with respect to health;

Context: Any setting (i.e., urban, suburban, rural).

The overarching objective of this review is to systematically search for, document, and analyze the inclusion of equity considerations and dimensions in smart cities research. To achieve this objective, we draw from Solanas et al.’s [34] sHealth concept definition and Trencher’s [9] outline of the smart city 2.0 paradigm and define human health-oriented smart city interventions as “any initiative, policy, promotion, program, or strategy that is conducted primarily in service of citizen needs (e.g., not a technological device simulation or health economic impact assessment). Said approaches can include the provision of a health service (e.g., substance abuse treatment), promotion of a health topic (e.g., mental health), a preventive strategy (e.g., physical activity), or targeting a clearly defined health outcome (e.g., improving mental health). Approaches are facilitated via the use of any urban context-aware network, sensing infrastructure, or other related technology”.

### 3.3. Search Strategy

Given the controversial history of “smart cities”, it is important to recognize the term’s inconsistent use as a synonymous expression for several related but distinct concepts over recent decades [63]. Commonly used interchangeable terms include “digital cities”, which have otherwise been defined as municipalities that make available for the public comprehensive, web-based representations of several functions or services [64]; “sustainable cities”, a term that has been used to suggest the utilization of various technologies in furtherance of mitigating CO_2_ emissions, to more efficiently produce energy and to enhance building efficiencies [65]; “intelligent cities”, a concept that delineates particularly innovative areas that seek to improve their communication and knowledge management via accessing the creative capital of local residents, institutions of knowledge creation, and digital infrastructure [66]; and “smart communities”, a term that proposes communities, neighborhoods, and regions, and their residents, organizations, and governing institutions employ information technologies for transformative purposes [67]. While we acknowledge the distinct definitions applied to each of these terms (and others, such as Internet of Things—IOT), we also understand that authors have used these terms synonymously when describing the interventions of interest in this review. As noted above, we are interested in identifying technological interventions in cities, which we define as “any initiative, policy, promotion, program, or strategy that is conducted primarily in service of citizen needs (e.g., not a technological device simulation or economic impact assessment)” [9]. To account for the synonymous (and inconsistent) use of terms to describe cities in which these types of interventions exist, we, therefore, use multiple terms in our search strategy to ensure full coverage of the interventions of interest.

With the help of a University of Waterloo librarian, the search strategy outlined two important conceptual categories. Variations of each concept (smart cities and public health) were discussed and developed by the authors, informed by previous systematic reviews [63]. Specific terms and phrases were truncated as necessary resulting in the following search strategy:


**Concept  Key Words**


[*Smart Cities*] “digital cit*” OR “green cit*” OR “information cit*” OR “intelligent cit*” OR “knowledge cit*” OR “smart cit*” OR “sustainable cit*” OR “ubiquitous cit*” OR “virtual cit*” OR “wired cit*” OR “internet of things” OR “IOT”

AND

[*Public Health*] “community health” OR “disease prevention” OR “electronic health” OR “ehealth” OR “epidemiology” OR “mobile health” OR “mhealth” OR “population health” OR “public health” OR “smart health” OR “sHealth” OR “wellbeing” OR “well-being” OR “well being”

The electronic databases used in this search needed to draw content from the fields of behavioral science, geography, public health, and urban planning. Due to having to incorporate this variety of research fields, the search strategy was carried out in five interdisciplinary databases: the Cumulative Index to Nursing and Allied Health Literature (CINAHL), Psychological Information Database (PsycINFO), the PubMed database from the National Center for Biotechnology Information, Elsevier’s database Scopus, and Web of Science. The database searches used to identify and document the articles presented in this review were current as of May 2019.

### 3.4. Eligibility Criteria

To be included in this review, all articles needed to satisfy five specific criteria. These criteria stipulated each study needed to have: (1) included a smart city approach; (2) contained some description or outline of the intervention design, approach, characteristics or features, implementation, evaluation, and outcomes (i.e., not a commentary or theory manuscript); (3) conducted real-world primary quantitative, qualitative, or mixed methods research (i.e., not grey literature, review, computer simulation, or feasibility report); (4) focused on some type of public health intervention; (5) measured a human health outcome either at the individual or community level (e.g., smart city housing impact on mental health, not on air filtering system efficiency); and (6) been written in English. There were no geographical or publishing time limits for where and when a given research article was conducted.

### 3.5. Study Selection and Review Process

Initial searches of the five databases yielded 3219 title results (see Figure 1). Specific returns from each database were 68 from CINAHL, 41 from PsycINFO, 227 from PubMed, 1969 from SCOPUS, and 914 from Web of Science. Screening for duplicates removed 923 articles. Title searches removed an additional 1058 titles, while the vetting of abstracts resulted in 1029 more records being excluded. During these initial screening steps, only titles and abstracts that were clearly and obviously not related to our research question were removed. In other words, any articles with an ambiguous title were kept for the of screening abstracts. Similarly, during abstract screening, any articles deemed as questionable with respect to our inclusion criteria were kept for the full-text scan. Full-text evaluations of the outstanding 209 articles concluded with another 196 papers being excluded. The most common justifications for the exclusion of papers during the full-text assessment phase included studies only running simulations of technologies, articles conducting evaluations of health technologies with no health outcome (e.g., usability of a technology, passive commuter game apps), manuscripts detailing health infrastructure with no clear human health outcome (e.g., cloud-based hospital management systems), texts not providing adequate descriptions of the health intervention, and studies presenting proof-of-concepts or blueprints. After full-text evaluations were completed, we conducted reference list searches of both included (13) and excluded articles, resulting in the addition of 15 more studies being added. Ultimately, 28 articles were retained and evaluated in the present review.

### 3.6. Data Extraction

The Cochrane PROGRESS-Plus equity tool (referring to (P) place of residence, (R) race, (O) occupation, (G) gender, (R) religion, (E) education, (S) socio-economic status, and (S) social capital) guided the extraction and thematic analysis of this review [60]. The tool outlines several essential social and personal characteristics that stratify the influences of and eventual health outcomes of individuals in different groups. When conducting the search, we defined equity as a measurement, analytical, or dialectical focus on systematic disparities in health between groups [68], and specifically coded for the tool’s prescribed characteristics, which included: place of residence (i.e., specific issues related to rural, urban, inner city, urban slums, remote communities); race, ethnicity, culture, or language (i.e., distinct background characteristics); occupation (i.e., occupation-specific risks and exposures); gender or sex; religion (i.e., group affiliation); education (i.e., individual education level attained); SES; and social capital (i.e., social relationships and networks). We also coded for personal characteristics (e.g., disability, age), features of relationships (e.g., smoking family members), and time-dependent relationships [61,69]. We documented and synthesized findings related to each of these equity characteristics in this review.

All extracted data from the articles included in this review are available in Appendix A. For each article, background information (i.e., author(s), year, sample information, study location, study design, and primary outcome) was extracted first. Data related to each equity characteristic were subsequently extracted by article section. Mention of specific frameworks or theories supporting an equity approach were collected from the introduction section of each article. Next, the methods section of each article was evaluated to record any equity characteristics relating to eligibility criteria, recruitment methods, research samples, and analysis methods. Results sections were then reviewed, with any data relating to the reporting of specific equity characteristics and their implications being collected. Lastly, data were extracted on any discussions pertaining to the relevant equity characteristics of the article.

### 3.7. Quality Assessment and Risk of Bias in Individual Studies

With this review including both quantitative and qualitative primary research studies, two specific quality assessments (QA) were used to judge the caliber of included articles. Two separate reviewers (A.B. and M.V.) reviewed both sets of articles, compared their evaluations, and negotiated the final scores for each manuscript. In the case of articles that used mixed methods, we graded each article based on how the outcome of interest (i.e., health outcome, or in the event of multiple health outcomes, whichever had the more substantial analysis) was assessed. For instance, if the outcome of interest was quantitatively evaluated while other outcomes were qualitatively appraised, we scored the article according to the quantitative tool.

With respect to the quantitative studies, we utilized the effective public health practice project (EPHPP) tool [70] (see Table 1). The EPHPP tool was selected due to its specific design for public health initiatives and use in previously published systemic reviews examining similar topics, such as mHealth interventions [71]. Following the EPHPP guidelines, the two reviewers compared scores to resolve any grading discrepancies and generate a global score, which was useful in accounting for inequalities in grading that arose during the individual assessments. Our QA examination of the quantitative articles found only three of the 21 studies to have strong global ratings. This finding is likely due to a number of factors. The studies generally had poor blinding components, which we construed as missing information, and therefore graded as weak. Selection bias was similarly poor, with most articles failing to provide details regarding the target population and the percentage of individuals who agreed to participate. Conversely, we observed that the studies had strong methods of data collection as scales and measures of outcomes were generally consistently reported. Overall, due to the EPHPP requiring only one weak rating out of six possible components to include it in the moderate global rating category, it is likely that the tool’s grading structure contributed to the frequency of moderate ratings (12 of 21 quantitative studies).

For the qualitative articles, we applied the National Institute for Health and Care Excellence’s (NICE) quality appraisal checklist tool for qualitative studies [72] (see Table 2). The NICE tool was used for similar reasons to those of the EPHPP, namely its specific design for public health topics and use in reviews that have examined health issues [73]. Similar to the EPHPP process, two reviewers compared scores to resolve any grading discrepancies and generate a global score based on negotiations of any variance of grading between the two individual raters. The NICE QA examination of the qualitative articles resulted in no articles being graded as providing the highest level of evidence (++). Ratings of the evaluations reflect that these articles did not typically have rigorous qualitative data analysis, and therefore likely contained biased or incomplete findings. While the research aims and study designs were often clear and detailed, the specifics regarding recruitment and data collection strategies tended to be scarce. Overall assessments of the included studies generally indicated that some or only few criteria were satisfactorily reported. Many of the mixed methods articles likely could not devote the space needed to satisfy all the criteria for the specific QA tools, resulting in lower scores than may have been possible had the articles been longer. Additionally, lower scores may also be a consequence of our including conference papers that had not gone through peer review.

## 4. Results

### 4.1. General Characteristics of Included Articles

General characteristics for each study can be found in Table 3. Importantly, when conducting the extractions, we did not consider reporting specific sample demographics (e.g., sex, gender, ethnicity) as a requisite piece of evidence for an article containing an equity focus. While there were no search date criteria applied in this review, the earliest noted evaluation being conducted on smart city interventions targeting human health and well-being was in 2013. Among the included studies, the majority (89%) of articles were based in Australia, Europe (particularly the United Kingdom), and the United States. The only article to incorporate and discuss a guiding theory for their intervention and analyses were Paredes et al. [74], who noted the use of the theory of implicit interaction. The heterogeneity of interventions with respect to strategies, tools, methods, aims, and required resources was vast. Findings related to each specific equity characteristic are discussed below.

### 4.2. Place of Residence

Equity considerations pertaining to place of residence were among the most pervasive throughout the included articles. While place of residence was a feature of analysis in one Pokemon Go study [75], it was largely a main focus or fundamental aspect in many smart city interventions. Incorporation of place-based considerations was notable for serving as a central concept in the outcome(s), concept, design, or evaluative processes of numerous smart city interventions. Within this category of place of residence, however, two distinct themes were present: (1) natural environments and their related specific features; and (2) commuting and commuter situations.

Place of residence as it related to concerns dealing with the natural environment and its specific features was often built into interventions as one of the explicitly stated research designs and outcomes. Specifically, evaluated location-based or time–space concepts included mental health outcomes based on exposure to different natural features (e.g., trees, birds, water) in urban built environments [76], citizens’ social interactions with nature in urban green areas [77], mood and mindfulness benefits derived from time spent in local outdoor or natural environments [78], wellness benefits of various natural environments [79], and well-being improvements generated from time in urban green spaces [80]. In this same vein, natural environment place of residence considerations were also observed more in the delivery of a smartphone application that promoted place-specific weather tracking and information updates designed for asthma management [81].

Secondly, there was a strong presence of place of residence considerations in the form of commuting-based interventions. Similarly, these considerations were central in the concepts, designs, and evaluations of the included articles. Examinations of context-orientated outcomes were observed in the evaluations of exercise opportunities for car commuters in traffic situations [82], commuters’ stress and perceived problems during trips to work as monitored by a travel companion [83], commuting stress in different simulated situations (e.g., city conditions, highway conditions) that drivers are commonly exposed to [84], commuters’ mindfulness during the course of trips [85], and commuter support via exposure to different themes (e.g., natural environments, urban scenes) based on a mood-sensing steering wheel [86]. Additionally, in commuting activities, lighting systems for pedestrians were investigated, assessing their impacts in numerous areas, such as urban (non-residential), leisure, residential, and traffic or commuting places [74].

### 4.3. Race, Ethnicity, Culture, Language

None of the included articles described an intervention specifically designed for or focused on different race, ethnicity, culture, or language backgrounds. While several studies reported race or ethnicity within their description of sample characteristics and data analysis and two Pokemon Go studies adjusted for race or ethnicity [75,87], there was no further consideration of these characteristics in the studies.

### 4.4. Occupation

The inclusion of occupation considerations were also comparatively minimal. In two evaluations, occupational considerations did have a principal role. First, an evaluation of a phone application sought to investigate ways to alleviate health caregivers’ challenges when caring for elderly patients [88]. Secondly, the discussion of a passive sensor-oriented toolkit designed to track meaningful daily activities, such as exercise and housework, touched on how the intervention could be of use to caregivers of individuals with Parkinson’s or similar diseases [89]. Outside of these instances, however, occupational status was only added as a covariate in the analysis of another intervention [76] that sought to examine the impact of natural environmental features on mental well-being.

### 4.5. Sex and Gender

Considerations of sex or gender in smart city health interventions were noted in a few articles. Many studies reported these characteristics in their sample overviews; however, more serious consideration of these specific features were limited to the data analyses. Most prominently, gender was discussed and noted as an important covariate in one of the analyses of the augmented reality Pokemon Go intervention, which promotes real world searches of video game characters appearing on a smartphone [90], while sex was included in three of the Pokemon Go evaluations [75,87,91]. Gender was also included in other work as an important factor of their evaluations, such as with the Urban Mind application, which documents the well-being benefits of exposure to natural features via tracking [76], and the Shmapped application, which collects well-being data from individual tracking in live field settings [80]. No interventions explicitly targeted or programmed for groups based on gender or sex characteristics or conducted a more in-depth analysis with these features (e.g., sex- and gender-based analysis).

### 4.6. Religion

No studies contained an intervention that was targeted for a particular religious affiliation. Religious affiliation or status was not a study priority, recruitment strategy, research design concept, or treated as an analytical covariate in any of the included articles.

### 4.7. Education

Similar to the gender and sex category, a few articles reported educational attainment as a part of their sample demographics. Only one intervention evaluation controlled for a measure of education level in their analysis [91], which found maternal education to be associated with lower sociability in a Pokemon Go study. Outside of these peripheral mentions, the determinant of individual education level was not a noted priority, recruitment strategy, or programmed feature in any of the smart city health interventions.

### 4.8. Socioeconomic Status

SES considerations were typically observed in the research designs and discussions of the implications of interventions. Regarding the former, low costs of public transit were explained as a part of the rationale in selecting passive commuters in one study [85], while SES was controlled for in the analysis of two of the Pokemon Go evaluations [75,91]. With respect to the latter, a few articles elaborated on concerns related to the affordability of their technology-based intervention [92] or the costs of app maintenance [80], while others discussed how, based on their evaluations, local governments need to ensure the economic accessibility of natural spaces for individuals of lower socio-economic backgrounds [78], while future research designs need to account [80] for low SES populations. No interventions were specifically targeted or programmed for low SES groups.

### 4.9. Social Capital

Social capital characteristics were among the more prevalent characteristics noted in the reviewed interventions. Most notably, a number of articles examined an outcome linked to social capital. Types of citizen social interactions taking place within urban green areas [77], contextual awareness and social interaction [92], social activity monitoring in healthcare settings [88], social isolation [93], and social cohesion and socially active lifestyles during the aging process [7] were all documented. More specific accounts of social inclusion were seen in some of the implications expounded on, specifically those explaining how smart technologies could potentially help to convey emotional states and better connect individuals [94], create more opportunities for people to make significant social connections with each other [95], and help build better support for caregivers [96].

### 4.10. Personal Characteristics

The consideration and inclusion of personal characteristics was common. Most prevalent was an evaluation examining or being tailored to groups based on personal characteristics that were either distinct health issues or age-related considerations. Particular health issues and conditions that were incorporated into the evaluations included stereotypical behaviors in children with autism [97], psychomotor development in children [98], and documented mental health issues [99]. Specific groups targeted by smart interventions, meanwhile, included those with dementia and Parkinson’s [89], college-aged students [99], patients with lung cancer [100], and people with asthma [81]. Other studies also incorporated into their analysis or discussions a variety of personal characteristic concerns, such as body mass index [75,90] or overweight [87], age [88,90,91,92], longevity and health or healthy aging [7], support for caregivers attending to those experiencing cognitive decline [96], and work-related stress [86].

## 5. Discussion

In this review, we identified and synthesized the equity approaches, characteristics, features, objectives, and outcomes of smart city interventions that were carried out to improve human health and well-being. Twenty-eight studies containing an evaluation of a smart city intervention which reported a human health outcome were systematically reviewed and assessed according to the Cochrane PROGRESS-Plus tool for their inclusion of different equity characteristics. While smart city concepts have historically been criticized for a lack of focus regarding social concerns [12,45,46,47], our review found that numerous equity issues and topics have been considered in smart city health interventions. In particular, place of residence, SES, social capital, and personal characteristics equity topics featured the most prominently throughout the reviewed interventions. Themes that emerged in the results, their implications for planning research and practice, limitations of this review, and future research directions are discussed below.

The equity concerns most commonly considered and included were place of residence, SES, social capital, and personal characteristics. Given the nature of this review examining smart city and health research topics, the agglomeration of these specific characteristics is perhaps somewhat predictable. Nevertheless, if such interventions are able to be scaled up, the inclusion and consideration of these particular equity concerns are encouraging developments for the prospects of achieving the global objective of health equity [38]. At a more local level, the documented emphasis of several interventions toward improving individual social capital (e.g., strong social networks) and better linking people with their local environments (e.g., green spaces) are similarly positive results, given that social interactions and active engagement are known correlates of building engaged and connected communities [101]. A continued emphasis on these equity aspects in the development, implementation, and dissemination of smart city interventions may help to create communities with more active and socially aware residents.

Equity characteristics related to race, ethnicity, culture, language, occupation, gender, sex, religion, and education, by contrast, were much more infrequently considered. While a few instances of inclusion regarding these equity features were recorded, there appears to be plenty of opportunity for their future inclusion. Cities and urban areas are increasingly taking on the majority of new immigrants in many countries such as Canada [102], and many newcomers may struggle to adapt and subsequently experience negative health outcomes during their move [103,104]. Interventions such as the ones involving the Shmapped [80] and Urban Mind [79] applications, which focus on connecting people with their local environments and provide opportunities for social interactions, could potentially be translated and targeted towards linguistic and cultural minority groups. Alternatively, with smartphone ownership greater than 80% in both developed and developing countries [105], these same interventions and others like them could also potentially be programmed to promote healthy behaviors for individuals with lower education levels who typically have poorer overall health [106].

Altogether, the reviewed smart city health interventions displayed a significant level of heterogeneity with respect to their strategies, tools, methods, aims, and required resources. Rather than one dominant set of concepts being observed across studies, numerous delivery methods including augmented reality [90], IoT frameworks [97], GPS tracking [76], mobile phone applications [77], and sensing technologies [86] were documented. Interestingly, within this heterogeneity, the nature of interventions and their evaluations appeared to reflect some of the notions of the smart city 2.0 paradigm, specifically the tenets of a bottom-up approach [14] and emphasis on more citizen-focused processes [20,21,22,23]. Citizen-focused processes are central to bottom-up approaches. As Calzada and Cobo [14] explain, they are reflected by an emphasis on the contextual influence of specific spaces, promotion of communitarian values over financial or private interests, and an intent in minimizing social and digital divides. Moreover, bottom-up approaches can promote important citizen-focused and democratic public processes, such as increased stakeholder involvement and municipal innovation movements that aim to foster the development of locally appropriate technologies [22]. While these observations may be an artifact of evaluators not possessing the resources required to conduct top-down, macro-level research, the targeted outcomes of interventions such as the monitoring of daily activities [7,96] and quality of life [100] and care [88] for specific populations, as well as emphasis on social and contextual influences in the designs of many studies, suggest an emphasis was placed on utilizing tech to improve individual quality of life and citizen needs rather than industry needs.

Most of the smart city intervention evaluations conducted were focused on personal and social spaces (e.g., urban green spaces) and places (e.g., home environments) rather than on workplaces. As evidenced by the dearth of details and examples present in the occupation section of the findings, there appears to be an opportunity for smart city technologies to be programmed for health in work- or occupation-specific environments. For example, Krome et al.’s [82] in-car fitness program and Pichlmair et al.’s [85] neck cushion and mobile app intervention could potentially be programmed to address the specific mental health and well-being of professional urban driving professions, such as rapid transport bus drivers, who experience some of the highest levels of work-related psychosocial risk [107,108]. Likewise, concepts such as Martindale et al.’s [95] IoT-connected plants, which focused on improving social connectedness, could potentially be expanded to specifically focus on office work environments, which have been found to induce stress among workers [109]. With the average worker in the United States spending 8.5 hours per day at their workplace [110], improvements for health in these spaces and environments could also produce lasting benefits for individuals. However, while discussing the different places where smart city interventions have been applied, it should be noted that cultural variance necessitates a variety of intervention approaches and applications. Political and social contexts across Africa, Asia, and Europe and their associated values, norms, and practices will factor heavily into determining which equity characteristics (e.g., occupation, religion) are targeted, the framing of these issues, and the intervention goals. Thus, it is important to reiterate that this review’s discussion largely takes place within a Western context due to the fact that the majority of reviewed studies were conducted in Western places.

Many evaluations of the interventions synthesized in this review seemed to still be in the early testing phases, and consequently did not feature robust evaluative designs or technologies that are widely or commercially available. There are few important implications stemming from this state of the literature. First, with many interventions seemingly in their nascent stages, it is unsurprising that the QAs conducted in this review outline many weaker or incomplete scores for the evaluations. Second, in light of these evaluative shortfalls, there is a need for more robust research designs to be used in the evaluation of future smart city health interventions. More rigorous evaluations can generate insights that provide clearer estimates of their effectiveness, public acceptance, and appropriate roles of smart city approaches in planning and other public sector processes. A greater number of studies including control groups, randomization techniques in their sample allocation strategies, and longitudinal research designs are recommended for future quantitative evaluations. Last, the documented lack of guiding theories used in the evaluations may lend some credence to the existing critique that smart cities lack a grounded understanding of the reality and dynamics of their social environments [51]. Theory-based research can elucidate important intrapersonal and interpersonal determinants affecting behavior and help improve the health effectiveness of interventions [111]. We suggest that inclusion of social and behavioral theory in future research, and especially in qualitative evaluations, to ensure rigorous methodologies are being applied in evaluations.

### 5.1. Implications for Practice and Policy

The findings of this review offer a few important implications for public health and planning. Planning and public health have a well-documented history of disconnection, despite having common historic origins [112]; however, recent calls have highlighted the potential of urban planning to address a number of critical health issues [1]. This review adds to these sentiments and specifically highlights that smart city health interventions can be adapted by local governments to promote strategies and educational messaging that promote both the autonomy and equity of individual citizens regarding their health outcomes. Having previously shown that societal disparities can drive the development of ensuing health issues, such as the growth of infectious diseases [42] and heightened violence and crime rates [43], this review also suggests that practitioners can use smart city technologies to mitigate the growth or entrenchment of damaging disparities in their communities. Likewise, local governments can also utilize smart city health technologies as preventive measures for at-risk or specific groups. For example, a recent randomized controlled trial assessment of a smartphone application programmed for smokers noted encouraging results with respect to smoking cessation [113]; local practitioners and governments can potentially incorporate context-aware features into similar interventions to more accurately address local population health issues or particular affected locations.

Regarding policy, theses such as the previously noted splintering urbanism posit that social cohesion and equity in urban areas are fundamentally being segmented by powerful coalitions that drive the economic liberalization of infrastructure [58]. Although the universality of this thesis has been critiqued [59], cross-cultural examples from Colombia [114], India [115], and South Africa [116] show that certain notions such as local policy featuring differentiation of services can be central drivers of inequities and fragmentation in urban areas. Above, we described different ways in which interventions targeting race, ethnicity, culture, language, occupation, gender, sex, religion, and education issues (e.g., smartphone promotions of health behaviors for low education groups) could potentially reduce urban health disparities. Having deliberated on this potential, findings from this review further support the idea that policy-makers should consider and legislate measures that seek to address some of the potential root causes of urban differentiation. For instance, a number of the reviewed interventions considered place of residence specifically with respect to well-being and mental health. Given that natural spaces (e.g., parks) can offer both mental health benefits [117] and opportunities for social interaction [118], policy-makers would be advised to prioritize and ensure the continued accessibility and quality of these public spaces for all groups in furtherance of promoting social cohesion and integration. Additionally, as others have observed compounding social and equity issues regarding the provision and connectivity of public transportation and housing supply for different groups [114], it is also critical for policy-makers to prioritize intersectoral equity objectives that target building diverse and sustainable communities. Neighborhoods that lack green open spaces, feature poor food retail options, and have a high density of alcohol and fast-food outlets can contribute to a myriad of disproportional deleterious health outcomes [119].

### 5.2. Limitations

When interpreting the findings of this review, there are a few important limitations to be aware of. First, we examined only select health-related smart city interventions that had been conducted with human subjects. Consequently, the review did not investigate the potential equitability of health interventions that have been carried out without human health outcomes (e.g., effectiveness of air filters, models examining different interventions’ emission reductions) but have the potential to affect human health. The review also only considered primary research with human participants, resulting in the exclusion of potential insights from secondary research overviews of interventions, descriptions of proposed initiatives, and feasibility studies of technology-based programs. Moreover, while we developed what we thought was a fairly broad definition of “smart city interventions’, we noticed the emergence of a few themes deriving from its application. Notably, we found that the definition resulted in the exclusion of hospital systems (e.g., cloud systems for data storage) and mobile primary health systems (e.g., patient appointment reminders), which may also have contained findings that could be important to note in the broader context of this conversation regarding technology and health equity. Finally, as previously noted, we recognize that the majority of included studies in this review were conducted in Western countries (e.g., Australia, Europe, United States), and thus reflect their research mandates, questions, priorities, and values. Despite this potential lack of cultural context, the aim of this review was to provide a baseline documentation of equity characteristics included in smart city health interventions; a baseline which we believe has been attained and can be used to help further larger aims, such as promoting global health equity.

### 5.3. Recommendations for Future Research

There are a few important areas for future research to investigate. Little is known about the potential of smart city interventions to promote the health and well-being of racial, ethnic, cultural, linguistic, and religious minority groups. This is important, as migration to urban areas will continue to intensify in the coming decades [4], which will result in socio-culturally and demographically diverse populations co-existing and co-locating in urban areas. Research regarding the use of smart city technologies targeting communities with lower educational levels is likewise scarce. More robust evaluations of health-based smart city interventions are also needed, particularly qualitative studies that can provide more in-depth insights pertaining to aspects such as the acceptance of technologies in targeted communities, desired intervention personalization features, and users’ perceived barriers. We also invite equity-based reviews to be conducted in other areas of planning and public sector governance, in order to further contextualize our findings by investigating how findings in related areas compare to those of this review. For instance, this review did not examine equity characteristics such as those related to health insurance, physically displaced groups, sustainability issues, or environmental concerns and exposures. Further investigation into these other areas and aspects can help to advance a more complete understanding of health equity across a variety of political, social, and technological contexts, which may ultimately support more holistic planning and public health practices.

## 6. Conclusions

The present paper provides a novel equity-based systematic review of smart city interventions targeting human health and well-being. Our equity-based analysis of the smart city health interventions suggests that place of residence, social capital, and personal characteristics are the most prominent considerations in such initiatives. There is potential for these interventions and future research to expand the equity considerations to occupation and workplace, education-based, and minority community health issues. Considering practical implications, we suggest that planners and other related practitioners utilize urban context-aware networks, sensing infrastructure, or other related technologies to promote health autonomy for individuals, mitigate local health disparities, and promote preventive measures for at-risk groups.

## Figures and Tables

**Figure 1 ijerph-17-02325-f001:**
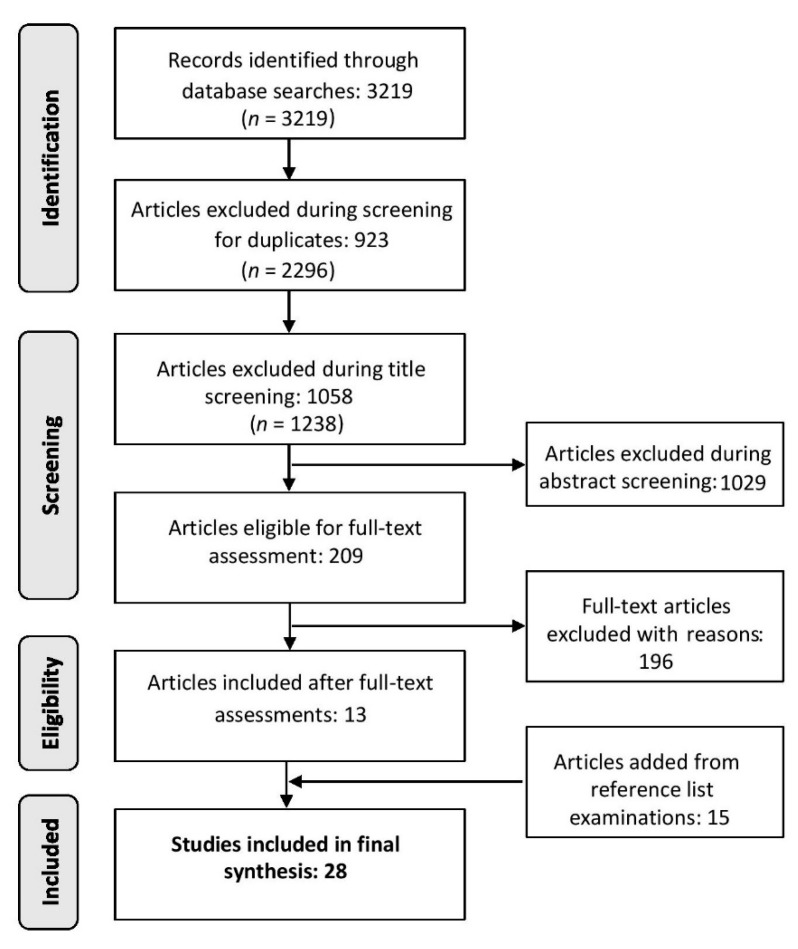
Study selection chart.

**Table 1 ijerph-17-02325-t001:** Quality Assessment of quantitative articles included in the systematic review (*n* = 21).

Author (Year)	Selection Bias	Study Design	Confounders	Blinding	Data Collection	Withdrawals and Dropouts	Global Rating
Althoff et al. (2016)	**	**	**	**	*	*N*/A	*
Amiri et al. (2017)	**	**	N/A	***	*	**	**
Bakolis et al. (2018)	**	**	N/A	***	*	**	**
Ferrara et al. (2018)	***	**	N/A	***	*	*N*/A	***
Frey et al. (2017)*	***	**	N/A	***	*	**	***
Gutiérriez Garcia et al. (2017)	**	**	N/A	***	*	**	**
Hamann et al. (2016)	**	**	***	**	*	**	**
Howe et al. (2016)	*	**	**	***	*	**	**
Isaac et al. (2018)	***	***	N/A	***	**	*N*/A	***
Lane et al. (2014)	**	*	**	***	**	**	**
MacKerron et al. (2013)	**	**	N/A	***	*	*N*/A	**
Martin et al. (2018)**	***	**	N/A	***	*	**	***
McEwan et al. (2019)***	*	*	*	**	*	**	*
Nef et al. (2015)	***	**	N/A	***	**	**	***
Nigg et al. (2017)	*	**	N/A	***	*	*N*/A	**
Paredes et al. (2016)****	**	**	N/A	***	**	**	**
Paredes et al. (2018)****	**	**	*	***	*	**	**
Park et al. (2019)	**	**	N/A	***	*	*	**
#Pichlmair et al. (2018)	**	**	N/A	***	*	**	**
Ruiz-Ariza et al. (2018)	**	*	*	**	*	*	*
Yang et al. (2019)	***	**	**	***	*	**	***

**Notes**: QA ool accessible via https://merst.ca/wp-content/uploads/2018/02/quality-assessment-tool_2010.pdf. Criteria Scale: * strong; ** moderate; *** weak. Global Rating System: * strong (no weak ratings); ** moderate (one weak rating); *** weak (two or more weak ratings). * Frey et al. (2018) used mixed methods and had multiple health outcomes, but evaluation was more quantitative than qualitative. ** Martin et al. (2015) used mixed methods but evaluated loneliness quantitatively. *** McEwan et al. (2019) used mixed methods but evaluated well-being quantitatively. Both Paredes et al. (2016) and (2018) used mixed methods and evaluated the health outcomes using both methods, however we deemed the quantitative analyses as being more substantial. # Pichlmair et al. (2018) used mixed methods and evaluated the health outcome using both methods, however we deemed the quantitative analysis as being more substantial. Abbreviations: *N*/A—not applicable; article used a study design with only one group, and therefore did not have any between group differences (confounders), or was retrospective (withdrawals and dropout).

**Table 2 ijerph-17-02325-t002:** Quality assessment of qualitative articles included in the systematic review (*n* = 7)

Author (Year)	Aims of the Research	Study Design	Recruitment and Data Collection	Data Analysis	Findings and Interpretation	Implications of Research	Overall Assessment of the Study
Davis et al. (2017) *	++	+	–	–	–	++	–
Klakegg et al. (2017)	++	++	–	–	–	+	–
Krome et al. (2017) **	++	+	+	–	–	+	+
Martindale et al. (2017)	++	++	–	+	–	+	+
Terken et al. (2013)	++	++	+	–	–	–	–
Tewell et al. (2019) ***	+	+	–	–	–	+	–
Trencher et al. (2017)	++	++	+	–	–	+	+

Notes: QA tool accessible via https://www.nice.org.uk/process/pmg4/chapter/appendix-h-quality-appraisal-checklist-qualitative-studies. Criteria Scale: ++ All or most of criteria fulfilled; + some criteria fulfilled; – few or no criteria fulfilled. Overall Assessment of the Study: ++ All or most of the criteria were fulfilled. Where they were not fulfilled, the conclusions of the study or review were thought to be very unlikely to alter the conclusions. + Some of the criteria were fulfilled. Those criteria that were not fulfilled or not adequately described were thought to be unlikely to alter the conclusions; – Few or no criteria fulfilled. The conclusions of the study were thought to be likely or very likely to alter the conclusions. * Davis et al. (2017) used mixed methods but the study evaluated the social connectedness outcome qualitatively. ** Krome et al. (2017) used mixed methods and evaluated the health outcome using both methods, however we deemed the qualitative analysis as more substantial. Tewell et al. (2019) used mixed methods but the outcome of interest was evaluated qualitatively.

**Table 3 ijerph-17-02325-t003:** General characteristics of included studies in systematic review (*n* = 28).

Author (Year)	Location *	Guiding Theory	Study Design	Health Outcome **	Equity Characteristic(s)	Intervention Description
Althoff et al. (2016)	United States	No	Retrospective Cohort Analytic	Physical activity	Gender/Sex Plus (P/C)	Pokemon Go app, augmented reality, and map tracking used to promote physical activity in real world searches.
Amiri et al. (2017)	United States	No	Cohort	Behavior detection	Plus (P/C)	WearSense, IoT framework with sensing capabilities in the form of stopwatches used to detect stereotypic behaviors in children with autism based on environmental surroundings.
Bakolis et al. (2018)	United Kingdom	No	Cohort	Mental well-being	Place of Residence Occupation Gender/Sex	Urban Mind app, smartphone-based tool that tracked exposure to natural features within the built environment and their impacts on mental well-being.
Davis et al. (2017)	Italy	No	MM Cohort	Social connectedness	Socio-economic Status Social Capital Plus (P/C)	IoT and ambient-assisted living environments, effects of ambient lighting configurations on cognitive performance, mood, and social connectedness.
Ferrara et al. (2018)	United Kingdom	No	Retrospective ITS	Well-being and nature interactions	Social Capital	Smartphone app featuring sensing capabilities, tracked citizen interactions with urban green areas and their impacts on well-being.
Frey et al. (2018)	France	No	MM Cohort	Breathing	Social Capital	Breeze wearable pendant, breath-sensing multi-modal biofeedback reported in real-time to assess breathing patterns.
Gutiérrez García et al. (2017)	Spain	No	Cohort	Psychomotor development	Plus (P/C)	Ubiquitous Detection Ecosystem to Care and Early Stimulation for Children with Developmental Disorders smart toy, stackable cubes equipped with sensors used to detect delays in psychomotor development in children in real environments (e.g., home, school).
Hamann et al. (2016)	United Kingdom	No	Cohort Analytic	Well-being	Place of Residence SES	Rewild Your Life online intervention, online program that promoted spending time in local nature to improve mood, well-being, meaning in life, and mindfulness.
Howe et al. (2016)	United States	No	Cohort Analytic	Physical activity	Place of Residence Race, Gender/Sex, SES, Plus (P/C)	Pokemon Go app, augmented reality and map tracking used to promote physical activity in real-world searches.
Isaac et al. (2018) ***	Australia	No	Cross-sectional **	Asthma	Place of Residence Plus (P/C)	Smartphone app that incorporated IoT features and real-time data on local environmental triggers (e.g., temperature, humidity) to inform asthma management.
Klakegg et al. (2017)	Australia	No	Cohort	Well-being	Occupation Social Capital Plus (P/C)	Mobile app which utilized sensors (“pervasive sensing approach”) to enhance care service for older adults by raising staff awareness of daily needs and routines.
Krome et al. (2017)	Australia	No	MM Cohort	Motivation for contextual exercise	Place of Residence	AutoGym, an in-car fitness program (mini-exercise bike linked to car speed utilizing sensors) run in a simulated rush hour driving scenario to promote physical exertion.
Lane et al. (2014) ****	United States	No	Cohort Analytic	Well-being	Social Capital	BeWell + app, runs on off-the-shelf sensor-enabled smartphones and was used to promote the adoption of healthy behavior (e.g., sleep patterns) via user feedback.
Mackerron et al. (2013)	United Kingdom	No	Retrospective ITS	Well-being	Place of Residence	Mappiness app, satellite positioning (GPS) was used to track participants and investigate momentary well-being when participants were in different environments.
Martin et al. (2018)	United Kingdom	No	MM Cohort	Stress	Place of Residence	Traeddy, an embedded technology augmented teddy bear (paired with an app) positioned as a well-being companion was used to inform car commuters about traffic situations and reduce stress.
Martindale et al. (2017)	United Kingdom	No	Cohort	Well-being	Social Capital	Connected Plants, examination of the potential of small-scale plants that incorporated IoT systems and collected personal data to promote health and wellbeing.
McEwan et al. (2019)	United Kingdom	No	Controlled Clinical Trial	Well-being	Place of Residence Gender/Sex SES	Shmapped app, smartphone app that used GPS to track participants and promote engaging in “geonarratives” to evaluate the impact of urban green space design on personal well-being.
Nef et al. (2015)	Switzerland	No	Cohort	Activities of daily living	Social Capital Plus (P/C)	Passive infrared sensors were installed in a smart apartment to detect and recognize eight different activities of daily living (e.g., cooking, sleeping, eating).
Nigg et al. (2017)	United States	No	Retrospective Cohort	Physical activity	Race/Ethnicity/Culture/Language Gender/Sex Plus (P/C)	Pokemon Go app, augmented reality and map tracking used to promote physical activity in real-world searches.
Paredes et al. (2016)	United States	Theory of implicit interaction	Cohort	Stress	Place of Residence	IoT interactive urban lights system, sensors used to respond to pedestrian traffic and designed to increase positive affect.
Paredes et al. (2018)	United States	No	MM Cohort Analytic	Breathing rate (i.e., stress)	Place of Residence	Physiological sensors (electrocardiogram, breathing rate harness, electrodermal activity bracelet) were used to assess reductions in drivers’ stress in simulated commuting environments.
Park et al. (2019)	South Korea	No	Cohort	Quality of Life	Plus (P/C)	Smart Aftercare app, an IoT wearable device connected with the app and other tools were used to assess the quality of life in patients with advanced lung cancer.
Pichlmair et al. (2018)	Germany	No	Cohort	Mindfulness	Place of Residence SES	Pen-Pen, a multi-component design which included the combination of a neck-cushion, a mobile app (which included GPS tracking), and a multi-modal feedback loop to improve mindfulness while commuting.
Ruiz-Ariza et al. (2018)	Spain	No	Randomized Controlled Trial	Emotional intelligence	Gender/Sex Education SES Plus (P/C)	Pokemon Go app, augmented reality and map tracking used to promote physical activity in real-world searches.
Terken et al. (2013)	Netherlands	No	Cohort	Stress	Place of Residence Plus (P/C)	In-car system that utilized a mood-sensing steering wheel and interactive in-car environment (i.e., images and sounds of a simulated environment) to assess mood and stress while commuting.
Tewell et al. (2019)	United Kingdom	No	MM Cohort	Meaningful activities	Occupation Plus (P/C)	Toolkit containing passive sensors used to assist individuals affected by dementia and Parkinson’s disease by monitoring meaningful activities in different home environments.
Trencher et al. (2017)	Japan	No	Cohort	Lifestyle activities	Social Capital Plus (P/C)	Multiple interventions carried out with wearable information communication technology devices, programs focused on assessing and monitoring daily activities (e.g., sleeping, walking).
Yang et al. (2019)	China	No	Cohort Analytic	Depression	Plus (P/C)	IoT structured wearable social sensing platform (wireless sensing technology used to connect with wearable devices, mobile phones, and server databases) used to assess mental state.

Notes: * In cases where no location for the study was explicitly mentioned, we used the location of the first author or location of where ethics were approved. ** We only list one health outcome in this chart, as that was the requirement in our inclusion criteria, however a number of studies report multiple health outcomes. *** Isaac et al. (2018) performed a single cross-sectional assessment of an asthma app. **** For Lane et al. (2018), the first author was affiliated with Microsoft in China, however nine of the other ten authors were affiliated with American institutions, thus we gave the United States as the location. Abbreviations: IoT—Internet of Things; ITS—interrupted time series; MM— mixed methods; P/C—personal characteristic(s).

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
