# Peer review of "Smart City and High-Tech Urban Interventions Targeting Human Health: An Equity-Focused Systematic Review"

_ijerph, 2020, doi:10.3390/ijerph17072325_

Round 1
Reviewer 1 Report
The article is well organized, it includes an appropiate state of the art, methodology, analisys and results and conclusions.
It is acceptable.
Reviewer 2 Report
Dear authors,
I am addressing some comments in the next lines:
- abstract: clarify the aim, since equity is a word with multiple meanings
- introduction: please complete it by describing the structure of the following sections
- section 2: your description of citizens' (and other actors') participation may be enriched by recalling Living Labs as contexts favouring a creative participation. Various papers address this topic
- section 2.1: please clarify how some services include health. Transportation for instance, I can spot a link, but it should be evident.
- section 2.2: missing equity in smart cities has been referred to even as splintering urbanism, with reference to multiple aspects and not just health. Please integrate it
- section 2.3: the justification of the study is weak. Why this methodology and this tool? Which are the advantages and the limitations depending on them? Are they new in this field?
- I would move section 2.3 and 2.4 in section 3
- section 3.1: the keywords you used are very different one another and they do not lead to smart city. Several papers stress the difference between i.e., digital and smart city, sustainable and smart city, and so on. This is really problematic in your manuscript
- if an eligibility criterion is 'include a smart city approach' how can 'sustainable city'-based papers/chapters be considered? Same applies to digital city
- Screening: is it really possible to exclude a paper/chapter just by reading the title?
- I would disagree on 'selection bias being poor' because your selection process is very subjective and in some cases arguable as in the previous comment
- discussion: the elements you considered as not affecting equity concerns may depend on local features, while I have a feeling you thought of them as poorly relevant in general
- section 5.1: there are not enough guidelines offered to policy-makers, neither to the web of actors shaping smart city projects
Reviewer 3 Report
The work maintains, in general terms, an adequate scientific level and its object of study I value it as appropriate, in line with the thematic and title of the publication. I also consider the originality of this proposal to be a strong point, both in terms of the subject matter and the hypotheses under study.
The measured use of bibliographic citations is appreciated, focusing in general on sufficiently contrasted and relevant works. The structure followed conforms to the usual points and the suggested order for a scientific article and, with the points to be corrected below detailed, the work has an adequate wording and syntax, respecting the grammatical correction rules
Questions of method and content for review paper:
This is a study that addresses an issue of current relevance how the Smart Cities are, their new socioeconomic dimension and impact relating it to the principles of Human Health sustainability.
The justification for the case study is very appropriate.
The paper should be present some figures or chart
Reviewer 4 Report
The review presents an equity-based analysis of smart city interventions directing to improve human health and well-being. It might be a topic of interest to both the planners and other related practitioners, but there are still some problems to be settled.
- As for the materials and methods part, the study selection chart presents a clear process from identification, screening, eligibility to included. Specifically in the quality assessment part, two quality assessments were used to judge the caliber of included articles. The method used works very well for examining smart city interventions; on the other hand, this is a well-established method, and the present research is a direct synthesis of this method without new contribution in methodological research, the authors may discuss more on the choice of two tools (EPHPP guidelines and NICE checklist).
- In the results section, the material was properly organized and some points of view presented are also very interesting. However, the results could be described concisely among the most pervasive throughout the included articles, e.g. giving examples of how smart technologies could potentially help to convey emotional states and better connect individuals.
- In the discussion section, findings, implications, limitations and future research directions were discussed orderly. As for the themes that most commonly considered in the results, place of residence, SES, social capital, and personal characteristics were analyzed rather few, mainly focusing on its potential possibilities. The authors might focus more on:1)whether these considerations are general equity characteristics, 2) if possible, add some counter-argument on ‘the inclusion and consideration of these particular equity concerns is encouraging development for the prospects of achieving the global objective of health equity’.
Reviewer 5 Report
The use of the smart city 2.0 paradigm makes sense - the paper shows a good application of it. A precise definition of equity would be useful, including a distinction to related concepts such as equality and justice. This could influence the assessment of whether and to what extent the studies have taken into account the concept of equity, especially when it goes beyond the mere indication of the use of certain sample demographics.Author Response
Please see the attachment.

Round 2
Reviewer 2 Report
Dear Authors,
thank you for the efforts in setting this new version. You dealt with most of the comments I addressed and in the first part of your manuscript things are set properly now, with clarifications in both the introduction and the literature review. In the remainder of the manuscript I observed several changes you made and I still have some concerns. I describe them in the following lines:
- section 3.3 (first and most relevant issue among the ones I cite in this list of recommendations): I read your justification about the different meaning of cities; I had read the contributions by Cocchia and Dameri & Rosenthal-S., as well as others dealing with the same topic. I appreciate your claim about acquiring a vision as wider as possible on 'new perspectives' on cities, but the search strategy is still problematic. Let me show you some definitions of these 'new' cities to claim what I meant and still mean: A) DIGITAL CITY 'The digital city is as a comprehensive, web-based representation, or reproduction, of several aspects or functions of a specific real city, open to non-experts'; B) SMART CITY 'smart city is currently being constructed as the solution to many urban problems, including crime, traffic congestion, inefficient services and economic stagnation'; c) SUSTAINABLE CITY 'a sustainable city is one in which 'its people and businesses continuously endeavour to improve their natural, built and cultural environments at neighbourhood and regional levels'. I selected three random definitions (I am not the author of any of these 3 definitions) and would invite you to compare them, because I know you are well aware of what they mean, but I feel you need to reason if they can be compared or not. I knew Cocchia stated 'digital city is often used like a synonymous of smart city', but - not to mention the other ones - what about sustainable cities? Therefore, I invite you to solve this issue; while reading your manuscript I thought you may perform an analysis just on smart cities as a first step. Then expand the analysis to the other concepts and claim why this has been necessary. By the way, this is a suggestion, your manuscript your way.
- To complement the comment above, look at your section 2: there is no mention at all of cities other than 'smart cities'. Therefore, the other (10) cities mentioned in the analysis come out of the blue, and the same applies to 'internet of things' and 'IOT'.
- One more comment on the issue above: your title says: 'smart cities'. This title can be misleading, if your idea is to expand the view on cities, an option should be to deal with 'transforming cities' or another tag you consider as suitable to different cities. In this case, section 2 should be enriched with a new section on the other cities As a reader I would be very disappointed in reading findings on sustainable cities, cyber cities, digital cities (these ones are just representations proposed by IT scholars or Architects), creative cities. Furthermore, sustainable cities were described/proposed in literature already at the end of 1990s, but recently they emerged again and with more and more relevance. The two approaches are a thousand miles far one from the other.
- Bias: I was perfectly aware what bias was used for in the previous version, thank you for clarifying. Anyhow, I used that reference to stress 2 problems affecting your search: the one mentioned above on concepts; the subjective selection of previous contributions. Now it is clearer when reading your response letter, but if one reads section 3.5 and looks at Figure 1, there are no clarifications on your selection process. You stated what you had done when selecting papers based on title and re-consider them if needed. Mention that, please. Same applies to the abstract-based selection.
- Discussion: the question from my previous review round was 'What about the local touch?'. More in detail, I meant a comment on the role of local culture as affecting the city transformation process was and is needed. I read your sub-section 4.3, but you should deal with culture - and other features describing the local context - some more. I can't image a smart city is based on the same approach in UK, Saudi Arabia, Japan, and Chile, just to mention 4 places with totally different cultures as well as with different studies on smart cities. For instance, transformed cities and cities built as smart are totally different.
- section 5.1: in your response letter you wrote 'certain notions like local policy featuring differentiation of services can be a central driver of inequities' and I definitely agree with you. Nevertheless, this statement is not mirrored in your discussion, as I referred to in the comment above. I saw what findings offer, but discussion should be more constructive, otherwise it would be a sort of 'Findings version 2'.
Kind regards
Reviewer 4 Report
All the questions were well answered.
Author Response
Thank you.
This manuscript is a resubmission of an earlier submission. The following is a list of the peer review reports and author responses from that submission.